# Biochemical and Structural Characterization of a Highly Glucose-Tolerant β-Glucosidase from the Termite *Reticulitermes perilucifugus*

**DOI:** 10.3390/ijms26073118

**Published:** 2025-03-28

**Authors:** Guotao Mao, Ming Song, Hao Li, Junhan Lin, Kai Wang, Qian Liu, Zengping Su, Hongsen Zhang, Lijuan Su, Hui Xie, Andong Song

**Affiliations:** 1College of Life Sciences, Henan Agricultural University, Zhengzhou 450046, China; maoguotao@henau.edu.cn (G.M.); songming@stu.henau.edu.cn (M.S.); 15993669760@163.com (H.L.); linjunhan@stu.henau.edu.cn (J.L.); kaiwang@gs.zzu.edu.cn (K.W.); 18703662970@163.com (Q.L.); suzengping@henau.edu.cn (Z.S.); hszhang@henau.edu.cn (H.Z.); sulijuan@henau.edu.cn (L.S.); 2The Key Laboratory of Enzyme Engineering of Agricultural Microbiology, Ministry of Agriculture and Rural Affairs, Henan Agricultural University, Zhengzhou 450046, China

**Keywords:** β-glucosidase, termite, glucose tolerance

## Abstract

The enzymatic hydrolysis of lignocellulose is often hindered by the glucose-mediated inhibition of β-glucosidases, a major bottleneck in industrial cellulose degradation. Identifying novel glucose-tolerant β-glucosidases is essential for enhancing saccharification efficiency. In this study, we cloned and heterologously expressed a novel β-glucosidase, RpBgl8, from the termite *Reticulitermes perilucifugus* in *Escherichia coli*. Sequence and structural analyses classified RpBgl8 as a glycoside hydrolase family 1 enzyme. The purified enzyme exhibited optimal activity at 45 °C and pH 7.0, with broad stability across pH 4.0–8.0. Notably, RpBgl8 demonstrated high tolerance to lignocellulose-derived inhibitors and organic solvents, maintaining 100% activity in 15% ethanol. Furthermore, RpBgl8 exhibited outstanding glucose tolerance, retaining 100% activity at 2.5 M glucose and 82% activity at 4.0 M glucose—outperforming most previously reported β-glucosidases. A structural analysis revealed a narrow, hydrophobic substrate pocket, with residue F124 at the glycone-binding site critical for minimizing glucose accumulation. The F124W mutation significantly reduced glucose tolerance, confirming that hydrophobic interactions at the active site mitigate inhibition. These findings establish RpBgl8 as a promising candidate for high-solid biomass processing and simultaneous saccharification and fermentation applications, highlighting termites as underexplored sources of biocatalysts with unique industrial potential.

## 1. Introduction

The sustainable production of biofuels and biochemicals from lignocellulosic biomass is pivotal to global efforts aimed at transitioning to renewable energy systems. Lignocellulose, the most abundant renewable carbon source in nature, consists primarily of cellulose (40–50%), hemicellulose (20–30%), and lignin (15–25%) [1]. The efficient enzymatic hydrolysis of cellulose, a linear polymer of β-1,4-linked glucose units, requires the coordinated action of three major cellulases: endo-1,4-β-D-glucanases (EC3.2.1.4), exo-1,4-β-D-glucanases (EC3.2.1.91), and β-D-glucosidase (EC 3.2.1.21) [2]. While endo- and exo-1,4-β-D-glucanases degrade cellulose into cellobiose, β-glucosidases hydrolyze cellobiose into glucose. This final step is essential not only for glucose release but also for alleviating the cellobiose-mediated feedback inhibition of upstream cellulases, thereby enhancing overall saccharification efficiency [3].

A major challenge in industrial cellulose hydrolysis is product inhibition, wherein the accumulation of glucose competitively inhibits β-glucosidase, reducing catalytic efficiency and requiring high enzyme loadings [4]. To overcome this limitation, glucose-tolerant β-glucosidases that maintain activity under elevated glucose concentrations have emerged as promising candidates for industrial applications, particularly in high-solid enzymatic hydrolysis and simultaneous saccharification and fermentation (SSF) systems [5]. Glycoside hydrolase family 1 (GH1) β-glucosidases are recognized for their high glucose tolerance, primarily attributed to the deep and narrow cavity that constrains the entrance of glucose into the substrate binding site [6]. Several glucose-tolerant β-glucosidases have been identified and characterized in recent studies, such as Bgl6, isolated from a metagenomic library of species from Turpan Depression [7,8], B8CYA8, from *Halothermothrix orenii*, GLU1, from *Microbacterium* sp., and LQ-BG5 derived from Hehua hot spring [9].

Termites, prolific lignocellulose consumers, produce a variety of cellulolytic enzymes [10,11,12]. Among these, termite-derived β-glucosidases have garnered interest due to their potential for high glucose tolerance. For instance, G1NKBG from *Neotermes koshunensis* retained over 100% activity under 0–0.8 M glucose [13], while wild-type (WT) MbmgBG1 and the R354K variant from *Macrotermes barneyi* maintained 40% and 100% activity, respectively, in 2.0 M glucose [14,15]. Despite this potential, termite-derived β-glucosidases remain underexplored compared to their fungal and bacterial homologs, leaving gaps in our understanding of their structure–function relationships and industrial applicability.

In this study, we identified and characterized a novel GH1 β-glucosidase, RpBgl8, cloned from the termite *Reticulitermes perilucifugus*. Recombinant RpBgl8, heterologously expressed in *Escherichia coli*, exhibited remarkable stability in the presence of lignocellulose-derived inhibitors, metal ions, and ethanol. Notably, RpBgl8 displayed high glucose tolerance, retaining full activity at 2.5 M glucose and 82% activity at 4.0 M glucose—outperforming most reported homologs. Structural and biochemical analyses provided mechanistic insights into the molecular basis of RpBgl8 glucose tolerance, offering a foundation for the rational engineering of robust β-glucosidases for industrial biorefineries.

## 2. Results and Discussion

### 2.1. Sequence and Structural Analyses of RpBgl8

A 1431 bp gene encoding the β-glucosidase RpBgl8 was cloned from *R. perilucifugus* (GenBank accession number: MN944396.1). The deduced RpBgl8 protein has a predicted molecular weight of 54.6 kDa. Sequence analysis using SMART identified a conserved GH1 domain, which was further confirmed by BLASTp analysis, classifying RpBgl8 as a GH1 β-glucosidase.

RpBgl8 shares 82% amino acid identity with the characterized β-glucosidase CfGlulC (GenBank: AEW67361.1) and 56% identity with the structurally resolved β-glucosidase G1NKBG (Figure 1). Multiple-sequence alignment revealed the presence of highly conserved catalytic residues, E168 in the NEPW motif and E375 in the TENG motif, which function as a general acid/base and a nucleophile, respectively, during the catalytic process of GH1 β-glucosidases (Figure 1a). Additionally, conserved residues, i.e., Q20, H123, F124, Y312, W417, E424, W425, and F433, were identified in RpBgl8, despite overall amino acid identity among homologs ranging from 35% to 56%.

The structure model of RpBgl8 generated using the AlphaFold server exhibited high quality (Appendix A) [16]. A structural analysis confirmed that RpBgl8 adopts the classical (β/α)_8_-barrel fold characteristic of GH1 enzymes, consistent with the predictions from the sequence analysis (Figure 1b). The catalytic residues E168 and E375 are positioned at the bottom of a deep substrate-binding pocket, which is enclosed by residues F124, Y312, W417, W425, and F433 on the C-terminal side of the barrel (Figure 1c). Glycoside hydrolases typically function through either a retaining or an inverting catalytic mechanism. In those using the retaining mechanism, the two catalytic residues are positioned approximately 5 Å apart, facilitating a two-step double-displacement reaction. In contrast, the inverting mechanism requires a greater separation (∼10 Å) to accommodate a water molecule for direct nucleophilic attack. In RpBgl8, the distance between E168 and E375 is about 5.4 Å, consistent with the catalytic mechanism characteristic of GH1 family β-glucosidases [17].

Collectively, the sequence and structural analyses confirmed that RpBgl8 is a GH1 β-glucosidase with conserved catalytic features and a substrate-binding architecture conducive to enzymatic activity.

### 2.2. Cloning, Purification, and Characterization of RpBgl8

As previously suggested, RpBgl8 was identified as a putative functional β-glucosidase. To verify its enzymatic activity, RpBgl8 was cloned, heterologously expressed in *E. coli*, and purified using Ni-NTA chromatography. The recombinant RpBgl8 carried a His-tag and a trigger factor (TF) tag at the N-terminus, and its molecular weight was consistent with the predicted size (Appendix A).

The enzymatic activity of purified RpBgl8 was subsequently characterized. As shown in Figure 2, RpBgl8 hydrolyzed 4-nitrophenyl-β-D-glucopyranoside (pNPG), a model substrate for β-glucosidases. The optimal pH for RpBgl8 activity was 7.0, with the enzyme retaining more than 80% of its maximum activity within a pH range of 6.0–8.0 (Figure 2a). Moreover, RpBgl8 exhibited high stability across different pH values, maintaining over 90% of its initial activity after 6 h of incubation at pH 4.0–8.0 (Figure 2b). RpBgl8 displayed maximal enzymatic activity at 45 °C. After incubation at 30 °C for 2 h, the enzyme retained over 95% of its initial activity. However, its thermal stability declined at higher temperatures (Figure 2d).

The kinetic parameters were determined using pNPG as the substrate at 37 °C, yielding a *K*_m_ value of 5.4 mM and a *k*_cat_ value of 10.2 min^−1^ (Figure 3). Although RpBgl8 exhibited hydrolytic activity toward pNPG and cellobiose (Appendix A), its catalytic efficiency was relatively low and will require further enhancement to effectively support cellulases in cellulose saccharification [18].

### 2.3. Effects of Chemicals on the Activity of RpBgl8

During lignocellulose pretreatment, various inhibitors, including 5-hydroxymethylfurfural (5-HMF) and furfural, are generated that can inhibit most cellulases [19,20]. The impact of 5-HMF and furfural on RpBgl8 activity was assessed (Figure 4a). RpBgl8 maintained 80–90% activity across furfural concentrations ranging from 0.25 to 2.0 g/L. Notably, RpBgl8 exhibited complete tolerance to 5-HMF, and at 2 g/L, 5-HMF enhanced RpBgl8 activity to 129%. Similarly, 5-HMF showed no inhibitory effect on the β-glucosidase from *Rasamsonia emersonii* [21].

The effects of metal ions and ethylenediaminetetraacetic acid (EDTA) on RpBgl8 activity are presented in Figure 4b. The presence of Na^+^, Mg^2+^, Ca^2+^, Ba^2+^, and EDTA did not significantly affect RpBgl8 activity. Interestingly, 10 mM Mn^2+^ enhanced RpBgl8 activity to 125%, a phenomenon also observed for the β-glucosidase from *Aeromonas* sp. HC11e-3 [22]. Meanwhile, the activity of the β-glucosidase CmGH1 from a deep-sea bacterium was inhibited to 64% by 10 mM Mn^2+^ [23]. Given the low thermostability of RpBgl8, the activation by Mn^2+^ may be attributed to its role in stabilizing the enzyme structure, as previously reported for the activation of the β-glucosidase BGL by Mn^2+^ [24].

Non-ionic surfactants such as Tween 20 have been shown to reduce the non-productive adsorption of cellulase enzymes on lignin, thereby improving cellulose hydrolysis [25]. As shown in Figure 4c, RpBgl8 retained 140% and 94% of its initial activity in the presence of 2.5% and 5% Tween 20, respectively. By comparison, the β-glucosidase CmGH1 completely lost its activity in the presence of 5% Tween 20 [23]. RpBgl8 exhibited 130% activity in the presence of 5% glycerol.

The effects of organic solvents on RpBgl8 activity were also evaluated. Remarkably, 2.5% methanol, butanol, and isopropanol increased RpBgl8 activity to 230%, 160%, and 190%, respectively (Figure 4c,d). RpBgl8 retained 51% activity in 20% isopropanol, whereas the β-glucosidase BGL1 from *Pichia etchellsii* lost 92% of its activity under similar conditions [26].

RpBgl8 exhibited high ethanol tolerance, maintaining more than 100% activity at ethanol concentrations ranging from 0% to 15% (Figure 4e). Notably, 5% ethanol enhanced RpBgl8 activity to 200%, and even at 20% ethanol, RpBgl8 retained 56% activity. In comparison, the ethanol-tolerant β-glucosidase BcBgl1A from *Bacillus cellulosilyticus* lost 50% of its activity in 15% ethanol [27], while Bg10 from a metagenomic library retained only 41% of its activity in 5.8% ethanol [28]. On one hand, ethanol typically reduces the stability of β-glucosidases [29]. On the other hand, ethanol can enhance enzyme activity by accelerating the release of the free enzyme from the glucosyl-enzyme intermediate via nucleophilic attack, compared to water [30]. These opposing effects may contribute to the apparent ethanol tolerance of RpBgl8. These findings highlighted the ethanol tolerance of RpBgl8, making it a promising candidate for second-generation biofuel production via SSF.

Overall, RpBgl8 exhibited high tolerance to lignocellulose-derived inhibitors, metal ions, and alcohols, underscoring its potential for industrial applications. However, the precise mechanism underlying the tolerance of RpBgl8 remains to be clarified through further biochemical and structural studies.

### 2.4. Glucose Tolerance of RpBgl8

During cellulose hydrolysis, glucose accumulates as the reaction progresses, leading to competitive inhibition of β-glucosidase. This inhibition reduces cellulose degradation efficiency, leading to cellobiose accumulation and subsequent feedback inhibition of cellulases [5]. Consequently, glucose tolerance is a critical parameter for evaluating β-glucosidase performance in industrial applications.

To assess the glucose tolerance of RpBgl8, its residual activity was measured in the presence of increasing glucose concentrations (0–4.0 M). Remarkably, RpBgl8 retained 100% activity at glucose concentrations up to 2.5 M and maintained 82% activity even at 4.0 M (Figure 5a). In contrast, the glucose-tolerant β-glucosidases Ks5A7 and Bgl6 lost 50% of their activity at 3.5 M and 1.5 M glucose, respectively, while the β-glucosidase O08324 from *Thermococcus* sp. still retained full activity at 4.0 M glucose [7,31,32] (Table 1).

These findings demonstrate that RpBgl8 exhibited high glucose tolerance, comparable to that of its most robust reported homologs. Its ability to function under high-glucose conditions underscores its potential for high-solid enzymatic cellulose hydrolysis, where elevated glucose concentrations typically impair enzymatic efficiency.

### 2.5. Molecular Basis for the High Glucose Tolerance of RpBgl8

The glucose tolerance of RpBgl8 was investigated through structural and biochemical analyses. Previous studies have reported that β-glucosidases with transglycosylation activity produce transglycosylation products in the presence of high glucose concentrations, contributing to their glucose tolerance [35]. However, thin-layer chromatography (TLC) of reaction mixtures containing pNPG and 0–1.0 M glucose confirmed the absence of transglycosylation products (Figure 5b), ruling out transglycosylation as a mechanism underlying RpBgl8’s glucose tolerance. Similarly, the glucose-tolerant Ks5A7 exhibited no transglycosylation activity in the presence of 5 mM pNPG and 250 mM glucose [36].

The structural analysis of glucose-tolerant and glucose-sensitive β–glucosidases has previously demonstrated a correlation between active-site accessibility and glucose tolerance [6]. To identify structural determinants of RpBgl8’s glucose tolerance, its substrate pocket was compared to that of glucose-tolerant enzymes (e.g., G1NKBG, 92% activity at 1.0 M glucose; HiBG, 100% activity at 0.45 M glucose) and glucose-sensitive homologs (e.g., TrBgl2, 50% activity at 0.05 M glucose) [13,34,37]. Structural superimposition revealed that the substrate pocket entrance of RpBgl8 closely resembled that of G1NKBG but was narrower than that of TrBgl2 (Figure 6). Quantitatively, the pocket volume of RpBgl8 (440 Å^3^) was slightly larger than that of HiBG (381 Å^3^) but significantly smaller than those of G1NKBG (643 Å^3^) and TrBgl2 (917 Å^3^), suggesting that a compact active-site geometry may limit glucose retention, thereby enhancing glucose tolerance.

The hydrophobicity of the substrate pocket has also been implicated in the glucose tolerance of β-glucosidases [33]. Further analysis highlighted the role of hydrophobicity in RpBgl8’s glucose tolerance. Sequence alignment of RpBgl8 with other glucose-tolerant β-glucosidases revealed a conserved Phe residue in the highly glucose-tolerant RpBgl8 (F124) and O08324 at the glycone-binding site, whereas other homologs contained a Trp at the equivalent position (Table 1). This distinction suggested that residue 124 plays a role in glucose tolerance. To investigate its contribution, F124 was mutated to Trp (F124W), which resulted in a 54.2% reduction of activity at 4.0 M glucose (Figure 5a). This result aligns with previous findings, where the reciprocal mutation (W122F) in B8CYA8 enhanced glucose tolerance [33]. Compared with Phe, the more hydrophilic Trp facilitated additional interaction with glucose via the indole ring [38], increasing local glucose retention and inhibition. In contrast, the hydrophobic Phe residue in the active site likely reduces glucose accumulation in the cavity, thereby enhancing glucose tolerance.

In summary, RpBgl8’s glucose tolerance is primarily attributed to its narrow, hydrophobic substrate-binding pocket, which minimizes glucose accumulation and mitigates feedback inhibition. These structural insights provide a foundation for rational enzyme engineering, where modulating active-site hydrophobicity could serve as a strategy to enhance enzymatic activity in industrial high-solid biomass processing.

## 3. Materials and Methods

### 3.1. Materials

The codon-optimized synthetic gene encoding RpBgl8 was ordered from Genscript (Nanjing, China). pNPG, 5-HMF, and furfural were obtained from Sigma-Aldrich (Saint Louis, MO, USA). Tween 20, methanol, ethanol, butanol, and isopropanol were purchased from Aladdin Scientific Corporation (Shanghai, China). The Silica gel 60 F_254_ aluminum plates were obtained from Merch KGaA (Darmstadt, Germany). All chemicals were of analytical grade unless otherwise specified.

### 3.2. Cloning of RpBgl8

Total RNA was extracted from the hindguts of *R. perilucifugus* (collected in Xinyang, Henan Province, China) and reverse-transcribed into cDNA using random hexamer primers. A putative β-glucosidase gene (*RpBgl8*) was amplified from the cDNA based on the homologous β-glucosidase gene from *Coptotermes formosanus*, and sequenced. The gene was codon-optimized for heterologous expression in *E. coli* and synthesized. Then, the gene was amplified with the RpBgl8-NdeI-F and RpBgl8-XbaI-R primers (Table 2, and subsequently cloned into the pCold-TF vector using NdeI and XbaI restriction sites. The recombinant plasmid pCold-RpBgl8 was sequence-verified.

### 3.3. Site-Directed Mutagenesis of RpBgl8

The F124W variant of RpBgl8 was generated via site-directed mutagenesis using the DpnI digestion method [39]. The primers F124W-F and F124W-R with the desired mutation were designed and synthesized (Table 2). PCR amplification was performed using the plasmid pCold-RpBgl8 as a template. The resulting product was treated with DpnI for 4 h at 37 °C and transfected into DH5α competent cells. The plasmid pCold-RpBgl8-F124W was isolated and sequence-verified to confirm the mutation.

### 3.4. Expression and Purification of Recombinant RpBgl8

The confirmed recombinant plasmids were transfected into BL21 (DE3) cells for the expression of RpBgl8 WT and the F124W mutant. The transformed *E. coli* cells were cultured in the Luria–Bertani (LB) broth supplemented with 100 mg/L ampicillin at 37 °C. Recombinant protein expression was induced by 0.5 mM isopropyl-β-thiogalactopyranoside (IPTG) for 14 h at 16 °C when OD_600_ reached 0.8. All purification steps were carried out at 4 °C. The cells were harvested by centrifugation at 4000× *g* for 30 min. The pellets were resuspended in the lysis buffer (20 mM sodium phosphate pH 7.4, 300 mM NaCl), and homogenized using a JN-Mini homogenizer (JNBio, Guanzhou, China). The supernatant was collected through centrifugation at 18,000× *g* for 30 min and passed through a Ni-NTA affinity column pre-equilibrated with the lysis buffer. Then, the column was washed with wash buffer (20 mM sodium phosphate pH 7.4, 300 mM NaCl, and 80 mM imidazole). The protein was eluted with elution buffer (20 mM sodium phosphate pH 7.4, 100 mM NaCl, and 200 mM imidazole). The eluted protein was concentrated and desalted with a 50 kDa cutoff concentrator. The purity and molecular mass of RpBgl8 was determined using 10% SDS-PAGE. The concentration of RpBgl8 was determined via the Bradford method.

### 3.5. Enzyme Activity and Stability Assays

The activity of RpBgl8 was assayed with pNPG as a substrate. The reaction mixture was prepared with appropriately diluted enzyme solutions and 5 mM pNPG in 20 mM phosphate buffer pH 7.0. The reaction was incubated at 37 °C for 10 min and then terminated by adding 50 μL of 1.0 M Na_2_CO_3_. The concentration of p-nitrophenol (pNP) was determined by measuring the absorbance at 408 nm using a SpectraMax M2e Microplate Reader (Molecular Devices, Sunnyvale, CA, USA). One unit of enzyme activity was defined as the amount of enzyme required to release 1 μmol of pNP per minute.

The optimum pH for RpBgl8 was determined in different pH buffers such as citric acid–sodium citrate (pH 3.0–5.0), phosphate buffer (pH 6.0–8.0), Tris-HCl buffer (pH 8.0–9.0), and NaHCO_3_-NaOH buffer (pH 10.0–11.0) at 37 °C. pH stability was determined by measuring the residual activity of RpBgl8 after incubation at 4 °C for 6 h in different-pH buffers.

The optimum temperature was determined in phosphate buffer pH 7.0 at temperatures ranging from 30 to 70 °C. Thermostability was determined by measuring the residual activity of RpBgl8 after incubation at 30–50 °C in a pH 7.0 buffer for various time intervals.

The kinetic parameters (*K*_m_ and *k*_cat_) were determined by assaying the enzyme activity at 37 °C in 20 mM phosphate buffer (pH 7.0) with different concentrations of pNPG (1–20 mM). The kinetic parameters were calculated by fitting the data to the Michaelis–Menten equation using GraphPad Prism 8.

### 3.6. Effects of Chemicals on the Activity of RpBgl8c

To evaluate the effects of chemicals on RpBgl8 activity, different concentrations of metal ions, organic solvents, and lignocellulose-derived inhibitors were added to the reaction system. The activity of RpBgl8 toward 5 mM pNPG in phosphate buffer pH 7.0 containing lignocellulose-derived inhibitors (0.25–2.0 g/L 5-HMF and furfural), metal ions (10 mM Na^+^, Mg^2+^, Ca^2+^, Mn^2+^, Zn^2+^, Ba^2+^, and EDTA), or organic solvents (2.5% and 5% tween 20, methanol, butanol, and glycerol; 2.5–20% ethanol; and 2.5–25% isopropanol) was measured relative to that of a no-additive control.

### 3.7. Glucose Tolerance Analysis

The glucose tolerance of RpBgl8 was determined by measuring the activity of RpBgl8 in phosphate buffer pH 7.0 containing different concentrations of glucose at 37 °C.

The transglycosylation activity of RpBgl8 was analyzed by TLC. The reactions were performed with pNPG as a substrate at 37 °C and pH 7.0 in the presence of glucose (0.5 and 1.0 M) and then boiled for 10 min to inactivate the enzyme. The samples were applied to Silica gel 60 F_254_ plates and then developed with ammonia/H_2_O/isopropanol (3:1:6, *v*/*v*/*v*) as a solvent system. After drying, the products were visualized with the anisaldehyde–sulfuric acid reagent.

### 3.8. Sequence and Structure Analyses of RpBgl8

The domain architecture of RpBgl8 was analyzed with the SMART program (http://smart.embl-heidelberg.de, accessed on 18 March 2022). Multiple-sequence alignment was performed using Clustal Omega (https://www.ebi.ac.uk/jdispatcher/msa/clustalo, accessed on 18 March 2022) [18]. Homology modeling of RpBgl8 was conducted using the AlphaFold 3 server (https://alphafoldserver.com, accessed on 20 March 2025) [16]. The volumes of the ligand binding pockets were calculated with the POCASA 1.1 server (https://g6altair.sci.hokudai.ac.jp/g6/service/pocasa, accessed on 20 March 2025) [40]. The structures were analyzed using Pymol 3.1 [41].

### 3.9. Statistical Analysis

All experiments were performed at least three times. The data are presented as mean ± SD. Statistical analysis and graphing were performed using GraphPad Prism 8.0 (San Diego, CA, USA) and Microsoft Excel 2016 (Microsoft Corporation, Redmond, WA, USA).

## 4. Conclusions

The GH1 family β-glucosidase RpBgl8 was cloned from the termite *R. perilucifugus* and heterologously expressed in *E. coli.* The purified RpBgl8 exhibited high tolerance to metal ions, lignocellulose-derived inhibitors (5-HMF, furfural), and organic solvents, particularly ethanol—retaining 100% activity at 15% ethanol—which underscores its suitability for SSF processes. RpBgl8 also displayed outstanding glucose tolerance, maintaining 100% activity at 2.5 M glucose and 82% activity at 4.0 M glucose, surpassing most reported β-glucosidases and meeting the requirements for high-solid enzymatic cellulose hydrolysis. The identification of RpBgl8 as a robust glucose- and ethanol-tolerant enzyme underscores termites as an underexplored yet promising source of biocatalysts with significant industrial potential.

Structural and biochemical analyses revealed that RpBgl8’s narrow, hydrophobic substrate pocket reduces glucose sequestration, providing a mechanistic basis for its resistance to product inhibition. However, its limited thermostability and low catalytic efficiency pose significant challenges for industrial applications. In recent years, computational protein engineering, rational design, and in silico directed evolution have become popular strategies for enhancing enzyme thermostability and catalytic efficiency [42,43,44,45], offering a promising approach to improving RpBgl8’s potential for biomass conversion platforms.

## Figures and Tables

**Figure 1 ijms-26-03118-f001:**
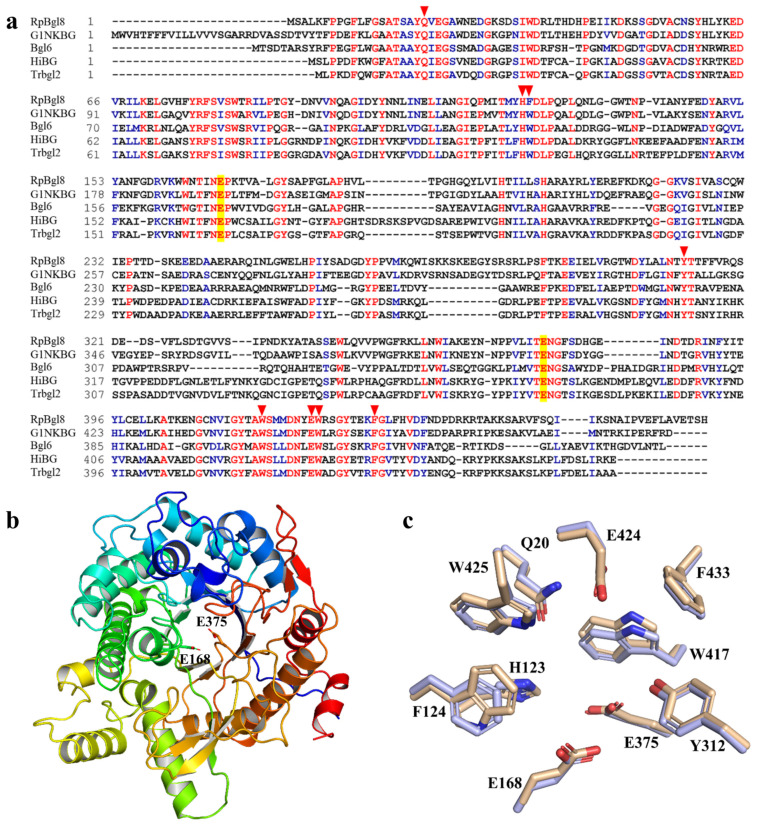
Sequence and structural analyses of RpBgl8. (**a**) Sequence alignment of RpBgl8 with G1NKBG, Bgl6, isolated from a metagenomic library of species from Turpan Depression (Uniprot: A0A0F7KKB7), HiBG, from *Humicola insolens* (O93784), TrBgl2, from *Trichoderma ressei* (O93785). Identical and similar residues are colored in red and blue, respectively. The acid/base and nucleophile residues are highlighted in yellow. The conserved residues forming the substrate pocket are labelled with triangles. (**b**) Structural model of RpBgl8. The acid/base and nucleophile residues are labelled and shown as sticks. (**c**) Superposition of conserved residues forming the substrate pocket of RpBgl8 (colored in light blue) and G1NKBG (colored in wheat, Protein Data Bank (PDB): 3vih).

**Figure 2 ijms-26-03118-f002:**
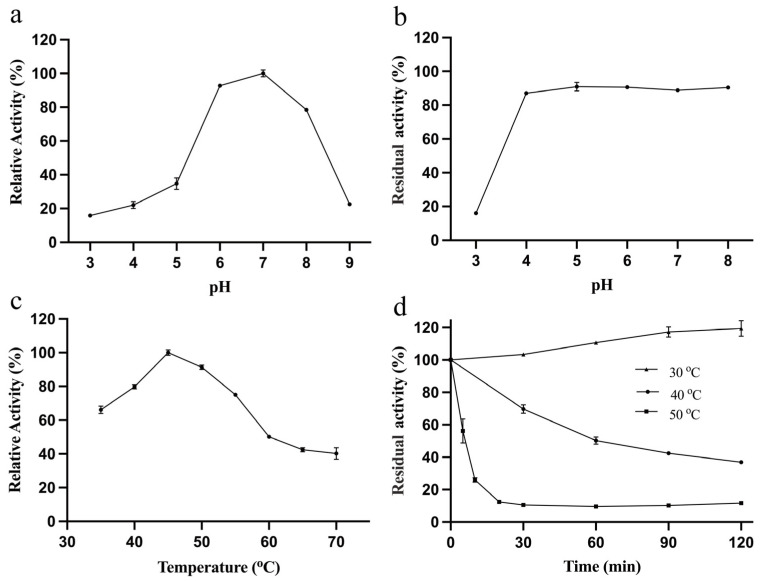
Effects of pH and temperature on the activity and stability of RpBgl8. The optimal pH (**a**) and temperature (**c**) for RpBgl8. Effects of pH (**b**) and temperature (**d**) on the stability of RpBgl8. pH stability was assessed by measuring the residual activity of RpBgl8 after incubation at 4 °C for 6 h in buffers of varying pH. Thermostability was evaluated by measuring the residual activity of RpBgl8 after incubation at 30–50 °C in a pH 7.0 buffer for different time intervals.

**Figure 3 ijms-26-03118-f003:**
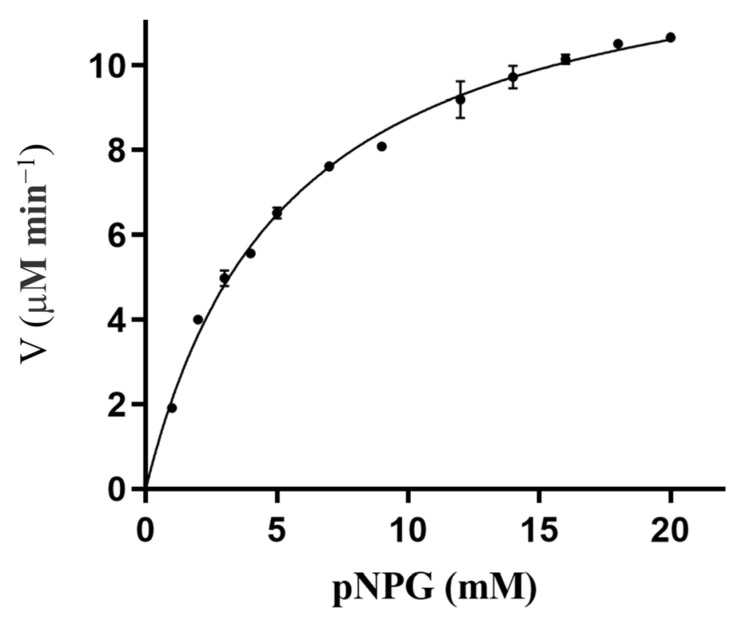
Kinetic analysis of RpBgl8 against 4-nitrophenyl-β-D-glucopyranoside (pNPG) at 37 °C.

**Figure 4 ijms-26-03118-f004:**
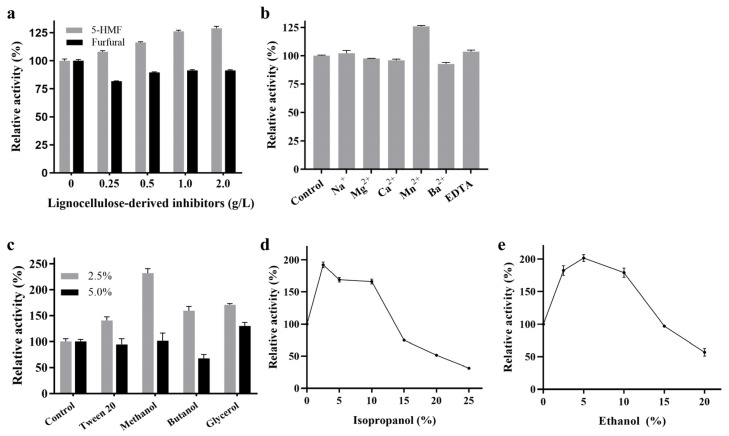
Effects of lignocellulose-derived inhibitors (**a**), metal ions (**b**), organic solvents (**c**), isopropanol (**d**), and ethanol (**e**) on the activity of RpBgl8. The activity of RpBgl8 was measured at 37 °C in 20 mM phosphate buffer pH 7.0 containing the specified additives and 5 mM pNPG. The final concentration of the tested metal ions and EDTA was 10 mM. EDTA: ethylenediaminetetraacetic acid; 5-HMF: 5-hydroxymethylfurfural.

**Figure 5 ijms-26-03118-f005:**
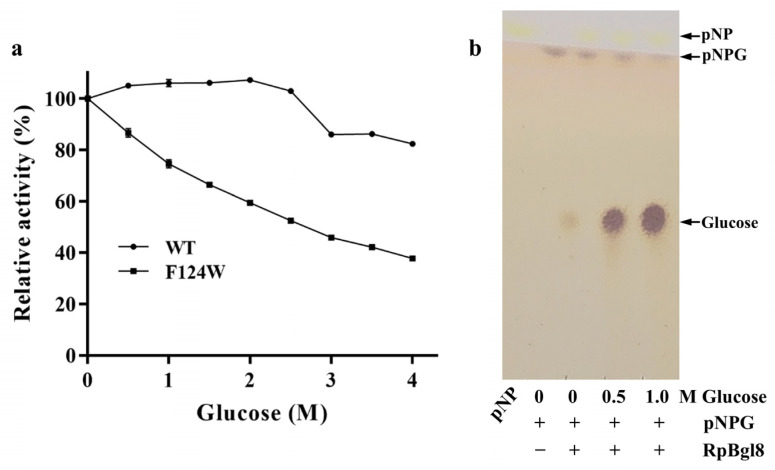
Glucose tolerance of RpBgl8. (**a**) The effect of glucose on the activity of RpBgl8. The activity of RpBgl8 in phosphate buffer pH 7.0 containing different concentrations of glucose was measured using 5 mM pNPG as the substrate at 37 °C. (**b**) The transglycosylation activity of RpBgl8. pNP: p-nitrophenol.

**Figure 6 ijms-26-03118-f006:**
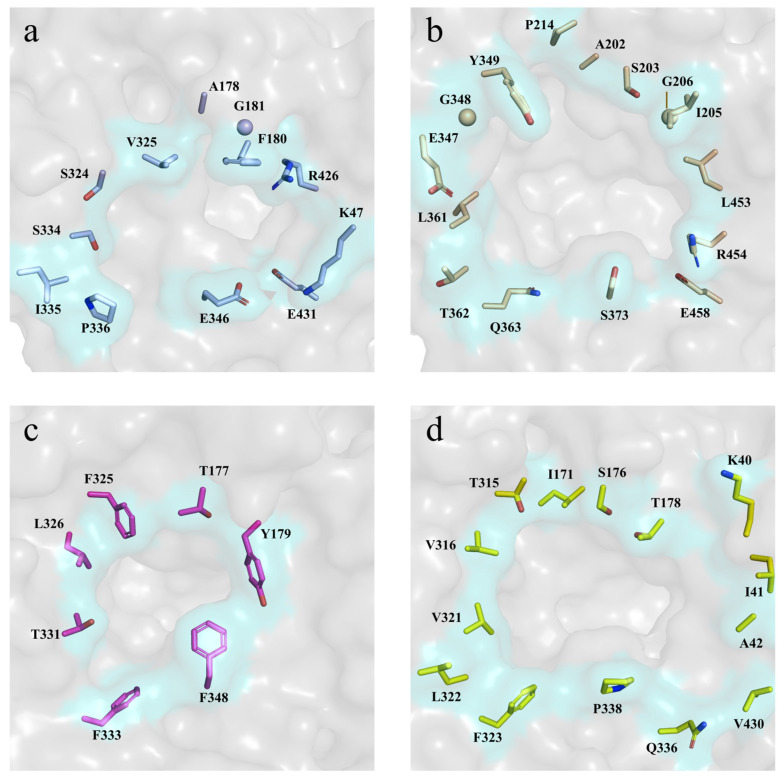
Entrances of the substrate pockets of RpBgl8 ((**a**), residues colored in light blue), G1NKBG ((**b**), wheat, PDB: 3ai0), HiBG ((**c**), light magenta, PDB: 4mdp), and TrBgl2 ((**d**), orange, PDB: 3ahy). The entrances of G1NKBG, HiBG, and TrBgl2 were superposed on the entrance of RpBgl8.

**Table 1 ijms-26-03118-t001:** The reported glucose-tolerant β-glucosidases.

β-Glucosidase	Glucose Tolerance	Residue 124 ^1^	Reference
RpBgl8	82% activity at 4.0 M glucose	F	This study
O08324	100% activity at 4.0 M glucose	F	[32]
Bgl6	50% activity at 3.5 M glucose	W	[7]
Ks5A7	50% activity at 1.35 M glucose	W	[31]
B8CYA8	110% activity at 1.5 M glucose	W	[33]
G1NKBG	92% activity at 1.0 M glucose	W	[13]
HiBG	100% activity at 0.45 M glucose	W	[34]

^1^ The residue number in the RpBgl8 sequence.

**Table 2 ijms-26-03118-t002:** The primers used to construct RpBgl8-overexpressing vectors.

Primer	Sequence
RpBgl8-NdeI-F	CGCCATATGAGCGCCCTGAAGTTCCC
RpBgl8-XbaI-R	TGCTCTAGAATGGCTGGTTTCCACGGCC
F124W-F	GTATCACTGGGACCTGCCGCAACCGCTG
F124W-R	GCAGGTCCCAGTGATACATGGTGATCATC

## Data Availability

The original contributions presented in this study are included in the article/Appendix A. Further inquiries can be directed to the corresponding author(s).

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
