# Peer review of "Biochemical and Structural Characterization of a Highly Glucose-Tolerant β-Glucosidase from the Termite Reticulitermes perilucifugus"

_ijms, 2025, doi:10.3390/ijms26073118_

Round 1
Reviewer 1 Report
Comments and Suggestions for Authors
Mao et al describe the production and characterization of novel glucose tolerant BG (RpBgl8). The enzyme is characterized in terms of the stability towards different compounds that are potentially present in lignocellulose hydrolysis and fermentation. Glucose tolerant BGs are of interest and the manuscript is reasonably well written. However, there are several issues to be addressed before this manuscript is suitable for publication.
Comments
In several places authors stress the potential of RpBgl8 in lignocellulose hydrolysis. I do agree that glucose tolerance is an advantage but this is only “one side of the story”. What is never discussed in this manuscript is the catalytic efficiency of RpBgl8. With pNPG as substrate the enzyme has kcat value of 10.2 1/min and Km is 5.4 mM (P4L126). This translates to the kcat/Km value of 31 1/M*s. RpBgl8 seems to be a poor enzyme. “Normal BGs” (i.e non-glucose-tolerant) have kcat/Km values for cellobiose in the order of 100,000 – 1,000,000 1/M*s and Ki for glucose around 1 mM (see e.g. Teugjas, H., Väljamäe, P. 2013 Selecting β-glucosidases to support cellulases in cellulose saccharification. Biotechnol. Biofuels 6, 105). One can estimate that in the presence of 10 M glucose, the BG with kcat/Km of 100,000 1/M*s and Ki of 1 mM performs as well as the BG with kcat/Km of 10 1/M*s and Ki of 10 M. My point is that high glucose tolerance (high Ki value) itself is not sufficient, the enzyme must also have reasonably high catalytic efficiency. Regarding this, authors should measure the catalytic efficiency of RpBgl8 with cellobiose substrate. The activity with pNPG model substrate is not relevant in the context of lignocellulose enzymatic hydrolysis.
High alcohol (ethanol, isopropanol methanol, etc, Fig 4) tolerance is confusing. First of all, it is not clear what is shown in Fig. 4. Is it the activity measured in the presence of indicated additives (Figure 4 leaves me with this impression, graphics says “relative activity”), or is it the residual activity measured after pre-incubation of RpBgl8 with indicated additives (Materials and methods indicates to this scenario, P10L302 says “residual activity”). Be precise here (in the legend to figure), what was the concentration of additives during pre-incubation and what was it during the activity measurements. What was the concentration of enzyme and substrate, pH, temperature? There is very little discussion in this manuscript. It just lists similar or different observations with other enzymes but never addresses mechanistic reasons. What is the reason for high (200 %) activity observed in the presence of ethanol (and other alcohols)? Is this because of the faster release of free enzyme from glucosyl-enzyme intermediate through the nucleophilic attack by alcohol (compared to water)? In this case the reaction product must be alcohol-glucoside and apparent alcohol tolerance is the mixed effect of increased activity and reduced stability of the enzyme protein.
The results about the absence of transglycosylation product in Fig 5B are not convincing. There is no pNPG substrate left in incubations with RpBgl8. It may be that translgycosylation product was there but it has been hydrolysed after depletion of pNPG substrate. Transglycosylation is usually under kinetic, not thermodynamic control.
Figure legends are virtually absent, there are just titles. Figure legends must contain sufficient information for understanding the figure without referring to the text.
Minor.
P1L26 – “glucose retention” is not easily understood.
P5L132 – Table 8… ?!
P6L166 – “exceptional ethanol tolerance” Is this a true ethanol tolerance or mixed effect of increased activity and moderate/low tolerance (see my major comment)?
P7L190 – “tolerance of”
P10L304 – Is this really a residual activity after incubation with glucose? Be more precise in describing your experiments. This is important.
P11L342 – There are other challenges besides low thermostability. See my major comment about catalytic efficiency.
Reviewer 2 Report
Comments and Suggestions for Authors
Mao et al. present biochemical and structural characterization of a glucose-tolerant beta-galactosidase. The findings are of general interest and the results might be useful in industrial cellulose degradation processes. I have no concerns related to the biochemical part of the research. However, there are some issues related to the structural part of the manuscript, and I believe once these are addressed, the manuscript may be suitable for publication.
- The structural model was obtained with somewhat obsolete software. Nowadays, the AlphaFold is the primary tool for in silico structure prediction. Also, there is no information on how the theoretical model was obtained. Although the authors indicate that such information is included in Figure S1, no SI file was attached to this submission.
- Page 4, lines 106-107: the authors claim that a 5.7A distance between E168 and E375 residues indicates a retaining mechanism for GH1. It should be explained more clearly (indicating reference 16 is insufficient).
- Page 5, line 132 - there is an “orphan” Table caption.
- Manganese ions enhance RpBgl8activity. Could the authors hypothesize a possible reason(s_ for that?
Round 2
Reviewer 1 Report
Comments and Suggestions for Authors
This paper is now acceptable.
Reviewer 2 Report
Comments and Suggestions for Authors
The authors improved the manuscript and provided the requested information and comments. Therefore, this work could be published in the IJMS.